# Male Stressed Mice Having Behavioral Control Exhibit Escalations in Dorsal Dentate Adult-Born Neurons and Spatial Memory

**DOI:** 10.3390/ijms24031983

**Published:** 2023-01-19

**Authors:** Li-Han Sun, Yi-Han Liao, Ya-Hsuan Chan, Anna E. Yu, Chun-Hsien Wu, Ing-Tiau Kuo, Lung Yu

**Affiliations:** 1Institute of Basic Medical Sciences, National Cheng Kung University College of Medicine, Tainan 70101, Taiwan; 2Department of Physiology, National Cheng Kung University College of Medicine, Tainan 70101, Taiwan; 3Chang Gung Memorial Hospital, Taoyuan 33305, Taiwan; 4Ditmanson Medical Foundation Chia-Yi Christian Hospital, Chiayi 600, Taiwan; 5Institute of Behavioral Medicine, National Cheng Kung University College of Medicine, Tainan 70101, Taiwan

**Keywords:** adult-born neuron, stress coping, corticosterone, operant conditioning, hippocampus

## Abstract

An escapable (ES)/inescapable stress (IS) paradigm was used to study whether behavioral control and repeated footshock stressors may affect adult neurogenesis and related cognitive function. Male stressed mice having behavioral control (ES) had a short-term escalation in dorsal dentate gyrus (DG) neurogenesis, while similarly stressed mice having no such control had unaltered neurogenesis as compared to control mice receiving no stressors. Paradoxically, ES and IS mice had comparable stress-induced corticosterone elevations throughout the stress regimen. Appetitive operant conditioning and forced running procedures were used to model learning and exercise effects in this escapable/inescapable paradigm. Further, conditioning and running procedures did not seem to affect the mice’s corticosterone or short-term neurogenesis. ES and IS mice did not show noticeable long-term changes in their dorsal DG neurogenesis, gliogenesis, local neuronal density, apoptosis, autophagic flux, or heterotypic stress responses. ES mice were found to have a greater number of previously labeled and functionally integrated DG neurons as compared to IS and control mice 6 weeks after the conclusion of the stressor regimen. Likewise, ES mice outperformed IS and non-stressed control mice for the first two, but not the remaining two, trials in the object location task. Compared to non-stressed controls, temozolomide-treated ES and IS mice having a lower number of dorsal DG 6-week-old neurons display poor performance in their object location working memory. These results, taken together, prompt us to conclude that repeated stressors, albeit their corticosterone secretion-stimulating effect, do not necessary affect adult dorsal DG neurogenesis. Moreover, stressed animals having behavioral control may display adult neurogenesis escalation in the dorsal DG. Furthermore, the number of 6-week-old and functionally-integrated neurons in the dorsal DG seems to confer the quality of spatial location working memory. Finally, these 6-week-old, adult-born neurons seem to contribute spatial location memory in a use-dependent manner.

## 1. Introduction

In the literature, the causative relationship between the psychological aspect of an individual’s control and its stress-buffering effects remains less explored. A lack of experimental support for such a causative relationship may be primarily due to the fact that few operative definitions are available for gauging such a sense of control. The earliest pioneers exploited immediate stressor termination—contingent upon animals’ objective-defined, and thus quantifiable, voluntary behavior—to model this psychological aspect of control [1,2,3]. As far as confounds associated with animals’ conditioning history and the physiological indices of choice, these investigators failed to determine the critical roles of the psychological aspect of control in modulating prospective stress-related pathology [2,3,4,5]. However, their use of “escapable/inescapable” stress appears to be an appropriate paradigm to study the impact of the psychological aspect of control (or behavioral control) on stress-elicited structural and functional plasticity [6,7,8].

Previously, we have demonstrated that one hour of intermittent footshock delivery may serve as a reliable stressor and cause rapid and robust increases in stressed mice’s serum corticosterone (CORT) levels [9]. In this study, we adopted a 10-day, daily, one-hour footshock stress regimen and an “escapable/inescapable” paradigm. “Escapable” footshock stress referred to mice experiencing the footshock stressor and behavioral control (ES), while “inescapable” stress referred to mice experiencing footshock without such control (IS). For each ES–IS dyad, ES, but not IS, mice were trained to immediately offset an ongoing footshock in the wake of their voluntary operant behavior [8]. As such, “ES” and their respective “IS” mice, who were also cage-mates, always underwent the same magnitudes of scheduled footshock in the same environment, with the former having behavioral (or the psychological aspect of) control and the latter having no such control. Importantly, 10 consecutive days of the footshock stressor regimen and a fixed ratio 1 schedule were used, in this regard, to allow ES mice to efficiently acquire the footshock-offsetting operant behavior, i.e., forward darting.

Acute stressors may render short-term neurogenesis decreases in the hippocampal dentate gyrus (DG) and related physiological aberrations in autophagy and apoptosis [10,11,12,13,14,15,16,17,18,19]. Stressor-induced rapid CORT secretion and heightened CORT levels may account for all these short-term changes [16,20,21,22]. Likewise, CORT-induced decreases in neurogenesis may be attributed to unbalanced autophagy and apoptosis in this regard [16,20,21,22]. However, whether repeated stressors, such as the 10-day footshock stressor regimen that we used in this study, may produce a long-term impact on DG neurogenesis, autophagy, or apoptosis remains elusive. Moreover, approximately 30% of adult-born neurons harbored in the dorsal DG are likely to survive at 6 weeks of age [23]. Adult-born neurons at this age may be mature and maximally integrated into the existing circuit to confer their physiological function [24,25,26,27,28,29,30,31]. Several lines of evidence have demonstrated that mature, adult-born dorsal DG neurons are paramount in their contributions to spatial learning and memory [32,33,34]. However, artifacts are suspected in these studies because their experimental methods may drastically change the number of neural stem cells in the DG [32,35]. In addition to these artifacts, conflicting results warrant reconciliation, because animals with adult-born neuron deficiencies may exhibit progressive progress in spatial learning tasks [32,34]. That is, it remains undetermined whether adult-born, mature, and functionally integrated dorsal DG neurons may participate in spatial working memory in a mature time- and use-dependent manner, as predicted by a school of theorists [33,36].

In this study, we first found that ES mice had a greater number of newly proliferative cells and proliferated neuroblasts in the dorsal, but not ventral, DG as compared to IS and non-stressed control mice 3 days after the conclusion of the 10-day footshock stressor regimen. Thus, we decided to assess whether operant conditioning (i.e., learning) and darting running (i.e., exercise) alone may be sufficient to induce these escalations. Moreover, we proposed the assessment of whether such a 10-day repeated stressor regimen may produce long-term effects on dorsal DG neuronal density, neurogenesis, gliogenesis, apoptosis, autophagic flux, and the animals’ heterotypic stress responses. Furthermore, the number of adult-born, functionally incorporated neurons in the dorsal DG was gauged in ES, IS, and control mice 6 weeks after the conclusion of the stressor regimen. Finally, we tested a hypothesis regarding whether the number of adult-born, dorsal DG neurons may confer the quality of the animals’ object location working memory in a neuron age- and use-dependent manner.

## 2. Results

### 2.1. ES Mice Had a Greater Number of Newly Proliferative Cells and Neuroblasts in Dorsal, but Not Ventral, DG

To assess the short-term effects of the repeated stressor and behavioral control on DG neurogenesis, 18 ES–IS dyads and their respective 18 non-stressed control mice were used. Their brain samples were obtained three days after the conclusion of the 10-day footshock stressor regimen. IS and control mice had comparable numbers of the BrdU- and BrdU/DCX-positive cells in their dorsal DG (Figure 1A,B). Surprisingly, ES mice exhibited a greater number of the BrdU- [F(2,27) = 49.99, *p* < 0.0001] and BrdU/DCX-positive [F(2,27) = 44.93, *p* < 0.0001] cells in their dorsal DG as compared with IS and non-stressed control mice (Figure 1A,B). Regardless of stress and behavioral control, mice displayed a comparable number of newly proliferative cells and proliferated neuroblasts in ventral DG three days after the conclusion of the 10-day regimen (Figure 1C,D). Although the BrdU labeling was completed throughout days 4–10 of the regimen, the footshock stressor regimen alone did not seem to affect cell proliferation or early neurogenesis in dorsal or ventral DG in IS mice. Importantly, these results seem to suggest that repeated stressors and behavioral control may exert a short-term escalating effect upon new cell proliferation and neurogenesis in the dorsal, but not ventral, DG.

### 2.2. Operant Conditioning and Forced Running Did Not Evoke CORT Secretion or Affect Short-Term Cell Proliferation or Early Neurogenesis in Dorsal DG

Using footshock and operant procedures, we demonstrated that 38 dyads’ averaged footshock-experienced durations progressively declined and reached an asymptote (approximately 3.5 s) of a descending valley on the third day of the 10-day regimen [F(9,333) = 32.83, *p* < 0.0001] (Appendix A). The implication of these results is that ES mice may reliably acquire footshock-offsetting behavioral control starting as early as day 3 of the 10-day regimen. Using Kruskal–Wallis tests followed by Dunn post hoc comparisons, ES and IS mice had indistinct serum corticosterone (CORT) levels, however, both were greater than those of the non-stressed controls 30 min after the conclusion of the 5th (x^2^ = 11.4, *p* = 0.0005) (*n* = 18) and 10th (x^2^ = 11.38, *p* = 0.0006) (*n* = 18) days of the regimen (Appendix A). The latter results suggest that ES and IS mice exhibit evident but comparable stress-induced CORT secretions at the second half (days 5–10) of the footshock regimen even though ES, but not IS, mice have behavioral control throughout.

To assess the plausible short-term impact of operant conditioning on dorsal DG cell proliferation and early neurogenesis, 10-day sucrose pellet-supported lever-pressing training, a form of appetitive operant conditioning, was employed in a group of mouse learners (*n* = 7) in one chamber. However, their respective non-learner yokes (*n* = 7) received simultaneous pellet delivery in another hooked chamber. The number representing the mice’s pellet gain was recorded, and a maximum of 60 pellets were available for each training day. Our results demonstrated that the learners’ average pellet gains reached the asymptote of a plateau on the third day of conditioning (Figure 2A). Nonetheless, 8 learners were found to have comparable CORT levels at 30 min after the conclusion of the 5th day of the training session as compared to 8 non-learner yokes (Figure 2B). Importantly, 7 learners and non-learner yokes had a comparable number of newly proliferative cells and proliferated neuroblasts in the dorsal DG (Figure 2C). These results suggest that operant conditioning alone does not seem to provoke changes in CORT secretion or early neurogenesis in the dorsal DG.

Moreover, we assessed whether forced running alone may elicit short-term escalations in proliferation and early neurogenesis in the dorsal DG. To this end, 10-day, daily, one-hour, tail base-grasping, attempt-induced darting was used among runners (*n* = 16), while it was not used among their respective non-runner yokes (*n* = 16). Runners and non-runner yokes were found to have not only indistinct CORT levels 30 min following the conclusion of the 5th day of this regimen (Figure 2D) but comparable numbers of BrdU- and BrdU/DCX-positive cells in the dorsal DG (Figure 2E). These results suggest that forced darting alone does not provoke changes in CORT secretion or early neurogenesis in the dorsal DG.

### 2.3. ES Mice Had a Greater Number of BrdU-Labeled Mature and Functionally Integrated Neurons in the Dorsal DG at 6 Weeks after the 10-Day Stressor Regimen

To gauge the differentiation and maturation of previously BrdU-labeled cells, BrdU and NeuN co-staining methods were used at 6 weeks after the conclusion of the stressor regimen. While 12 IS and control mice had a comparable number of BrdU/NeuN-positive cells, 12 ES mice had a greater number of BrdU/NeuN-positive [F(2,33) = 20.66, *p* < 0.0001] cells in dorsal DG as compared to IS and controls (Figure 3A,B). In contrast, three groups of mice (*n* = 24) had sparse and indistinct BrdU/GFAP-positive cells in this regard (Appendix A). To further test whether those BrdU-labeled mature neurons may be integrated into extant circuitry at this time point, kainic acid-induced seizures and Arc protein co-labeling methods were used. Eight ES mice displayed a greater number of BrdU/NeuN/Arc-positive cells in the dorsal DG as compared with the 8 non-stressed control and IS mice [F(2,21) = 22.76, *p* < 0.0001] (Figure 3C,D). The latter two groups of mice had an indistinct number of BrdU/NeuN/Arc-positive cells in the dorsal DG (Figure 3C,D). An implication of these results is that the majority of the 6-week-old, adult-born neurons are incorporated into the existing circuits in the dorsal DG.

### 2.4. The Repeated Stressor Regimen did Not Produce Long-Term Effects on Dorsal DG Mitosis, Neuronal Density, Apoptosis, Autophagic Flux, or Animals’ Heterotypic Stress Response

To assess the long-term effects of the 10-day repeated footshock stressor regimen and behavioral control on the dorsal DG micro-environment, mice’s brain tissues were collected 6 weeks after the conclusion of the 10-day footshock stressor regimen. Dorsal DG Ki-67-positive cell counts (*n* = 24) and local neuronal (NeuN-positive) density (*n* = 21) were comparable among ES, IS, and control mice (Figure 4A–D). Likewise, the repeated stressor regimen and behavioral control did not seem to affect the number of existing cells undergoing apoptotic death (*n* = 12) (Figure 4E,F) or local autophagic flux (including LC3II/LC3I and p62) (*n* = 30) (Figure 4G) 6 weeks after the conclusion of the stressor regimen. Repeated stressors may render long-lasting changes in both homotypic and heterotypic stress responses and related plasticity in many brain regions [38]. To test whether our repeated footshock stressor regimen may cause evident long-term stress response plasticity, a total of 36 mice underwent a novel stressor at 6 weeks after the conclusion of the 10-day regimen to reveal their heterotypic stress responses. Mice were subjected to an elevated circular platform (diameter = 1 m) under 1000-lux illumination and 70 dB of white noise for a total of 35 min. Thirty minutes after this stress exposure, the mice’s trunk blood was collected and their serum CORT assay was completed. We noticed that the ES–IS dyads and the respective non-stressed controls had comparable CORT levels (Figure 4H). An implication of the latter results is that the repeated footshock stressor regimen and behavioral control produce, at best, a minor long-term effect on animals’ heterotypic stress response. These results, taken together, suggest that repeated stressors and behavioral control produce negligible long-term effects on animals’ subsequent stress response and local mitosis, extant neuronal density, apoptosis, and autophagic flux in the dorsal DG.

### 2.5. ES Mice Outperformed IS and Control Mice in the Object Location Task on the First and Second Day of the Trial Starting 6 Weeks after the Conclusion of the 10-Day Stressor Regimen

To assess whether the 10-day footshock stressor regimen may exert a short-term, if any, impact on animals’ object location working memory, four versions were employed in a within-group, counterbalancing manner (one-quarter of the mice from each group received rectangular, regular hexagon, regular pentagon, and parallelogram arena, respectively) at 3 days after the conclusion of the stressor regimen (*n* = 36). We found that neither the 10-day stressor regimen nor behavioral control seemed to affect mouse object location working memory at this time point (Figure 5A). Using similar designs, the 10-day stressor regimen and behavioral control did not seem to affect mice’s object location working memory at 3 weeks after the conclusion of the regimen (*n* = 36) (Figure 5B). A seminal implication of these results is that young (less than 4 weeks of age or so), adult-born dorsal DG neurons do not seem to confer on spatial location working memory.

To prevent a repeated measure (i.e., memory) effect in object location performances, four versions were employed in a between- and within-group counterbalancing manner for 4 consecutive days of daily trials. The 12 ES mice demonstrated significantly greater recognition ratios compared to the 12 IS and non-stressed control mice on the first day of the trial at 6 weeks after the conclusion of the stressor regimen [F(2,33) = 7.277, *p* = 0.0024] (Figure 5C, upper left). Likewise, ES mice also outperformed their respective yoke, IS, mice, and control mice on the second day of the trial [F(2,33) = 5.174, *p* = 0.0111] (Figure 5C, upper right). Nonetheless, ES mice’s performance was indistinct from their respective IS and control mice in this recognition ratio on the third and fourth days of the trial (Figure 5C, bottom). Moreover, no correlation was noticed between ES mice’s decreases in footshock-terminating latency on the third day of the regimen and their averaged recognition ratios on the first and second days of the trial 6 weeks after the stressor regimen. The latter results suggest that the mice’s naïve learning seems to be loosely correlated with their spatial memory performances. Six weeks after the conclusion of the repeated stressor regimen, ES mice outperformed IS and control mice in object location working memory in the initial two, but not the two remaining, trials.

### 2.6. TMZ Treatment Deteriorated Mice’s Performance in Two Consecutive Days of Object Location Task 6 Weeks after the Conclusion of the 10-Day Stressor Regimen

To vigorously test the hypothesis that the number of 6-week-old, adult-born dorsal DG neurons may be associated with the animals’ performance of object location working memory, 12 ES and IS mice received the TMZ treatment protocol (see Appendix A, *n* = 12). Non-stressed control mice receiving an equal volume of saline (*n* = 12) or no injections (*n* = 9) served as controls. Six weeks after the conclusion of the stressor regimen, mice underwent two days of the trial for their object location working memory using systematic between- and within-group counterbalancing designs. A total of three saline injections did not seem to affect the mice’s object location working memory, the amount of exploration time, or the BrdU-labeled neuron number. Compared to both non-stressed control groups, ES and IS mice similarly exhibited object location working memory on the first and second day of trials [F(3,82) = 16.58, *p* < 0.0001] (Figure 6A, top). It is important to note that four groups of mice displayed indistinct amounts of time spent in object exploration during these two trials (Figure 6A, bottom). Moreover, non-stressed controls had a significantly greater number of BrdU-labeled, 6-week-old neurons in the dorsal DG as compared with TMZ-treated ES and IS mice [F(3,41) = 9.863, *p* < 0.0001] (Figure 6B). However, TMZ-treated ES mice had comparable numbers of (BrdU/NeuN)-labeled, 6-week-old neurons as compared with TMZ-treated IS mice in the dorsal DG (Figure 6B). These results support the hypothesis that the number of 6-week-old, adult-born dorsal DG neurons may confer the quality of animals’ object location working memory.

## 3. Discussion

Repeated presentation of the same stressor is unavoidable to reduce experimental animals’ surprise responses and surprise-stimulated CORT secretion [39]. Thus, the repeated presentation of the same scheduled footshock is suspected to progressively taper animals’ surprise. In support of this surprise-stimulated CORT notion, the ES–IS dyads’ CORT levels were found to be 3–4 times greater than non-stressed controls’ levels immediately after the 5th day of the stressor session. However, ES and IS dyads’ CORT levels were found to be less than 2 times greater than those of non-stressed controls after the 10th day of the stressor session. While psychosocial and physical stress have been known to induce reliable decreases in early neurogenesis [11,40,41], IS and non-stressed control mice had comparable numbers of newly proliferated cells and proliferative neuroblasts in the dorsal DG. Such unaltered short-term neurogenesis in IS mice’s dorsal DG may not be attributed to the surprise- and/or CORT- tapering effect. After all, stressed mice, regardless of ES and IS grouping, had significantly greater CORT secretion as compared to the non-stressed controls at the end of the 10-day regimen. Likewise, IS mice’s unaltered short-term neurogenesis in the dorsal DG is less likely to be due to the presence of a single ES throughout the stressor regimen [9].

To date, most, if not all, study results have shown that stressor-evoked CORT secretion is sufficient to yield decreases in DG neurogenesis [20,42,43]. Undergoing a robust stressor regimen and heightened CORT secretion, our IS mice showed unaltered short-term dorsal DG neurogenesis. These paradoxical findings suggest two possibilities. First, an anti-CORT effect, for maintaining homeostasis and/or directing allostasis, may be progressively developing following repeated stress-provoked CORT secretions. Stressor-induced oxytocin secretion may exert such an anti-CORT effect to prevent IS mice from the same stressor-induced decreases in short-term dorsal DG proliferation and neurogenesis [9,44]. Likewise, repeated footshock stressor-programmed lymphocyte and cytokine production may also serve as two candidates for endogenous, anti-CORT substrates [45]. Along with this possibility, one may predict IS mice’s ostensible reduction in short-term dorsal DG neurogenesis provided such anti-CORT substrate production is prevented throughout the day 4–10 BrdU-labeling and stressor regimen. Second, repeated stressor-provoked mitochondrial metabolism bias may boost the transition from quiescence to the activation of the neural stem cells and neuroblasts [46]. Further, fumarate, a Krebs cycle metabolite, may be regarded as a repeated stressor-biased molecular candidate in this regard [47]. As such, BrdU-labeled mitotic cells near the end of the 10-day regimen are expected to outnumber the labeled cells on day 4.

Since ES and IS dyads’ CORT levels were comparable, and both were greater than non-stressed controls’ at days 5 and 10 of the stressor regimen, it was obvious that behavioral control did not affect the stressor-stimulated CORT secretion. A previous study demonstrates that ES and IS animals may exhibit comparable CORT increases following a single footshock stressor session [6]. Our results extend their findings by showing that behavioral control does not seem to exert its effects on stressor-evoked neuro-endocrine acute responses even if repeated and/or long-term stressors are present. Under a stress condition, behavioral control seems to boost dorsal DG neurogenesis, very likely via mechanistic underpinnings emerging and prevailing after the rapid activation of the hypothalamic–pituitary–adrenal axis and brain CORT distribution.

ES mice had a greater number of newly proliferated cells and proliferative neuroblasts in the dorsal DG compared to their IS counterparts and non-stressed controls 3 days after the conclusion of the stressor regimen. In the literature, physical exercise, enriched environment, learning, and mating experiences have been documented for their potential neurogenesis-boosting effects [48]. To assess whether our ES mice’s early neurogenesis escalations may be due to their operant conditioning experiences, another operant conditioning method of 10-day, sucrose pellet-supported, lever pressing was used. Using this operant conditioning method, mouse learners had an immensely similar acquisition time point of “behavioral control” as the footshock-terminating ES mice. However, this pellet-supported, lever-pressing conditioning did not seem to affect early neurogenesis in learners’ dorsal DG, suggesting that ES mice’s conditioning experience alone plays a minor, at best, role in elevating their early neurogenesis in the dorsal DG. Likewise, we demonstrated that 10-day forced darting alone did not seem to affect runners’ early neurogenesis in the dorsal DG. These results, taken together, suggest forward darting exercise and operant learning are not sufficient to yield such early neurogenesis escalation in the dorsal DG. It was of special interest to note that neither learners nor runners showed elevated CORT levels 30 min after the conclusion of their 5th day of training regimens. However, footshock-terminating ES mice demonstrated significantly elevated CORT levels on the 5th day of training, and even so, at the end of their 10-day regimen. A contingency of the immediate termination of punishment (or the presentation of a reward) and preceding voluntary behavior was eminent in our ES, runner, and learner mice. Only ES mice’s behavioral control may adapt robust stress into a neurogenesis-invigorating stimulation and/or to exonerate metabolic balance from potentially stress-produced allostatic loads [8,49]. That is, cementing with tangible stress, behavioral control may elevate early neurogenesis in the dorsal DG.

ES mice appeared to have indistinct local physiological and metabolic conditions, including extant neuronal density, cell apoptotic death, and autophagic flux as compared with their IS counterparts in the dorsal DG 6 weeks after the stressor regimen. Moreover, our repeated footshock stressor regimen exerted a negligible long-term effect on dorsal DG mitosis, which was in parallel with previous findings [20]. While various forms of stress may affect dorsal DG neurogenesis, local aberrant mitochondrial autophagic flux and apoptotic death seem to be two leading factors mediating those stress effects on neurogenesis [18,50,51,52]. The implication of our results on mitosis, apoptosis, and autophagic flux is that the local microenvironment in the dorsal DG is resilient to the observable long-term effects of the 10-day stressor regimen [53]. As such, ES mice’s extant dorsal DG microenvironment is expected minorly contribute to their better performance in object location memory 6 weeks after the stressor regimen. Moreover, no correlation was noticed between ES mice’s lever-pressing latency (on the third day of the stressor regimen) and their averaged (the first two days of trials) recognition ratios for the object location task. That is, ES mice’s early progress in operant training, reflecting their naïve learning/memory, confers little, at best, on their prospective object location performances.

ES mice had a heightened number of BrdU-labeled, mature, and functionally integrated neurons in the dorsal DG 6 weeks after the conclusion of the stressor regimen. Both the existing and adult-born neurons in DG are susceptible to seizure-induced damage [54]. Nonetheless, our BrdU/NeuN/Arc-staining findings demonstrated that most, if not all, BrdU-labeled mature neurons seemed to be functionally incorporated into extant neuronal circuitry at this time point [36]. Moreover, ES mice outperformed their respective IS and non-stressed controls in the dorsal DG-related object location task [55] exclusively at 6 weeks, but not at 3 days or 3 weeks, after the stressor regimen. The latter findings are in parallel with recent electrophysiological findings documenting that the excitation of 6–8-week-old new mature DG neurons are actively involved in behavioral plasticity [31]. Per our results, ES mice’s greater number of 6-week-old, functionally-integrated neurons in the dorsal DG seem to correlate positively with their object location performance. This time-dependent correlation is very likely attributed to the maturation of the adult-born neurons and their exuberant integration into the existing EC-DG system at this time point [27,29].

While immature, adult-born granule cells are paramount in spatial memory formation and extinction [30,32,56], mature, adult-generated dentate neurons are also critical for spatial learning consolidation and retrieval [57,58,59]. These lines of evidence assist investigators with formulating a theoretical framework supporting the unique role of adult-born DG neurons in discriminating the most recent and/or ever-changing compound information encoding from remote and/or previous ones [36,60,61,62]. Compared to IS and non-stressed mice, ES mice with a greater number of 6-week-old neurons in the dorsal DG were found to outperform in object location working memory during the first two trials. However, the same ES mice exhibited comparable performance levels in such object location working memory as IS and non-stressed mice in the following two trials. These results suggest that 6-week-old, adult-born neurons in the dorsal DG seem to be salutary in boosting animals’ object location working memory in a use-dependent manner. It was important to note that the daily training chambers and the objects used were novel to our experimental mice across the four-day trials. Thus, discriminating or updating new (unfamiliar) cues/contexts from familiar ones is always required within and between adjoining trials. Conforming to the aforementioned theoretical framework, we empirically demonstrated that ES mice’s greater numbers of 6-week-old, adult-born neurons biased their good performances on the first two, but not the following two, trials. Their task performances, however, descended to the non-stressed controls’ level in the following two trials. A reasonable explanation for the latter results is that 6-week-old, adult-born neurons lose their new spatial working memory-boosting effects after they are used for a previous encoding purpose. An alternative, but not necessarily exclusive, explanation is that a greater number of 6-week-old, adult-born neurons fade in their spatial memory-boosting role after they are involved in spatial memory consolidation. Finally, it is also very likely that ES mice are less motivated to recruit those 6-week-old, adult-born neurons to form object location working memory as the loss of novelty progresses over the 4-day trials of the object location task. After all, four object location trials shared common features in novelty-driving exploration [63]. Thus, mice’s principle learning (or meta-learning) seems to be a potential factor to back up the last explanation. Nonetheless, this principle learning is less likely because non-stressed control mice do not display eminent, if any, alterations in these working memory performances across these 4-day trials.

While ES mice’s number of 6-week-old dorsal DG neurons was positively associated with their object location memory, their causative relationship remains an open issue. To tackle these issues, a TMZ treatment protocol was attempted for its effectiveness in preventing short-term neurogenesis escalation in ES mice. Both ES and IS mice, then, received the TMZ protocol to yield tangible decreases in the number of 6-week-old, adult-born neurons in the dorsal DG. Compared with the non-stressed controls receiving vehicle (saline) or no injections, both TMZ-treated ES and IS mice failed to perform reliable object location memory at 6 weeks after the stressor regimen. Likewise, non-stressed controls’ 6-week-old, adult-born dorsal DG neurons outnumbered those of the TMZ-treated ES and IS mice. Further, TMZ-treated ES mice had an indistinct number of 6-week-old dorsal DG neurons. The latter results suggest that a minimal number of 6-week-old dorsal DG neurons seems to be necessary to allow animals to perform intact object location working memory tasks. Local existing neurons older than 6 weeks of age, in contrast, seem to play, at best, a minor role in performing this object location working memory task.

In this report, we show that behavioral control cementing with repeated stressors may paradoxically enhance early neurogenesis in the dorsal DG. The number of 6-week-old mature neurons in the dorsal DG may correlate with animals’ object location working memory performances in a positive manner. Alzheimer’s disease at its early stage manifests with working memory issues and spatial disorientation [64]. In this regard, our findings may provide a mechanistic perspective in support of a well-known meta-analysis theory that independence activity training and cognitive stimulation seem to be useful interventions in retarding the progression of Alzheimer’s disease [65]. Through repeated training activity (i.e., a repeated stress-provoking situation), the subjective gain of behavioral control is anticipated to modify the training-related and psychological stress into invigorating challenges and, likewise, to enhance adult-born neurogenesis in the dorsal DG. Working memory regarding spatial direction, cues, and cognitive maps may be resilient to aging-related regression when an individual’s stress is joined with behavioral control [64]. Further translational study is warranted to test the efficacy of upgrading the magnitudes of both stress and respective behavioral control for retarding progressive cognitive declines in patients afflicted with Alzheimer’s disease.

## 4. Materials and Methods

### 4.1. Animals

To avoid the complex modulating effects of female sex hormones on neuron differentiation and survival [37], only male mice were used in this study. Eight-week-old C57BL/6 mice were obtained from the National Laboratory Animal Center (Tainan, Taiwan, ROC) and the NCKUCM Laboratory Animal Center (Tainan, Taiwan, ROC). Mice were group housed in plastic cages (4–5 per cage, 29 × 19 × 12 cm) in a temperature- and humidity-controlled colony room on a 12 h light/dark cycle with lights on at 07:00. All mice had access to tap water and Purina Mouse Chow (Richmond, IN, USA) *ad libitum*, unless otherwise mentioned. All experiments were conducted in temperature (23 ± 1 °C)- and humidity (70%)-controlled laboratories. A total of 562 mice were used in this study, and all experiments were performed in accordance with the National Institutes of Health Guide for the Care and Use of Laboratory Animals revised in 2011. All procedures were approved by the local Animal Care Committee at the National Cheng Kung University College of Medicine (NCKUCM No. 108160).

### 4.2. The 10-Day Footshock Stressor Regimen and Behavioral Control

The 10-day footshock stressor regimen was previously described [8]. In brief, this regimen consisted of daily (1 h session/day) footshock (a total of 60 pseudo-randomly-scheduled footshocks/hour, 0.5 mA each) delivery in a 24 cm long, trough-shape metal chamber [66] for 10 consecutive days (Figure 7). Cage mates with comparable weights were randomly assigned as ES, IS, and non-stressed control mice. Each dyad’s ES and IS mice received the 10-day, daily footshock stressor regimen, while their non-stressed controls received no such regimen. Throughout the footshock delivery regimen, each footshock was set to be on for 7 s continuously or subjected to immediate termination contingent upon ES, but not IS, mice’s forward darting behavior. Each ES and IS dyad underwent the same magnitudes of scheduled footshock (including the number and onset/offset timing of each footshock) throughout this 10-day stressor regimen. ES mice’s behavioral control referred to their performance of the footshock-terminating operant behavior.

### 4.3. Quantification of Short-Term Effects of Stress and Behavioral Control on DG Neurogenesis and their Long-Term Effects on the Local Micro-Environment

ES mice acquired reliable behavioral control on the third day of the footshock stressor regimen. We, thus, decided to use 7 consecutive injections (100 mg/kg/injection at 1-day intervals) of bromodeoxyuridine (BrdU; Sigma Chemical, St. Louis, MO, USA, Cat. No. B5002) to label newly proliferated DG cells starting on the fourth day of the regimen. ES, IS, and control mice received an intraperitoneal BrdU injection immediately prior to the daily session of the footshock regimen. Bromodeoxyuridine co-staining with doublecortin (DCX), a microtubule-associated protein in neuroblasts, methods were used to reveal DG early differentiated neuroblasts. Mice’s newly proliferated cells and proliferative neuroblasts in the dorsal and ventral DG were assessed 3 days after the conclusion of the 10-day stressor regimen. Mice were deeply anesthetized with sodium pentobarbital and transcardially perfused with ice-cold 0.1 M phosphate-buffered saline (PBS, pH adjusted to 7.4), followed by 4% paraformaldehyde in ice-cold 0.1 M PBS. Their brains were removed and postfixed in a 4% paraformaldehyde solution overnight at 4 °C and subsequently cryoprotected in a 30% sucrose solution for 48 h at 4 °C. Coronal sections at 20 μm thickness were made using a microtome (Thermo Fisher Scientific, Cleveland, OH, USA, Model: CryoStar NX50 OP). Brain slices were incubated in a 50% formamide/2xSSC (sodium chloride/sodium citrate) buffer for 2 h at 65 °C, rinsed with 2x SSC, and incubated in 2 N hydrochloric acid for 30 min at 37 °C. Slices were then rinsed with PBST (triton-100-containing PBS, 1%) buffer and incubated in a blocking buffer (BSA:240 g, goat serum:160 μL, sheep serum:160 μL in 8 mL of PBS) for 2 h at room temperature. The slices were then stained by mouse anti-BrdU (1:500, Millipore, Temecula, CA, USA, Cat. No. MAB4072) (Appendix A) for newly proliferated cells. For identifying newly differentiated neuroblasts, brain slices were co-immunostained with primary antibody (rabbit anti-DCX, 1:500, Cell Signaling, Danvers, MA, USA, Cat. No. 4604S) (Appendix A), incubated with FITC and TRITC conjugated (1:500, Jackson ImmunoResearch, West Grave, PA, USA, Cat. No. 515-545-062 and 111-585-045) secondary antibody (Appendix A), and imaged with an Olympus fluorescent microscope (Olympus, Tokyo, Japan, Model: IX71). Since staining was quantified in the subgranular zone (hilus not included) of the dorsal and ventral DG (bregma: −1.34 to −2.30 mm and −2.46 to −4.04 mm, respectively), an average of 50 and 36 coronal sections was obtained for each mouse. Using a stereological method, the total number of BrdU- and BrdU/DCX-positive cells in a series of every 7th section spaced at 120 μm was obtained and then divided by the slice selection ratio (i.e., 8/48 for the dorsal and 13/79 for the ventral DG). Exploiting z-stack methods to process the presence of fluorescence-positive cells, a composite z-stack image was first generated by composing 3–4 projection-compressed images using CellSens software (DP2-BSW ver.2.2, Olympus, Tokyo, Japan) for each 20 μm slice. On those composite z-stack images, staining positive spots were then counted by a rater blind to the grouping.

To assess the long-term impact of stress and behavioral control on dorsal DG cell proliferation, Ki-67 staining methods and the aforementioned stereological and z-stack microscopic methods were employed 6 weeks after the conclusion of the stressor regimen. Likewise, local neuronal density, apoptotic cells, and autophagic flux indicators (LC3II/LC3I and p62) were also assessed 6 weeks after the conclusion of the regimen [8,67]. Mouse brain slices containing dorsal DGs were stained by rabbit anti-Ki-67 (1:1000, Abcam, Cambridge, MA, USA, Cat. No.ab15580) (Appendix A) in a blocking buffer containing 3% BSA and 3% goat serum in PBS and incubated with Alexa Fluor 594-conjugated goat anti-rabbit secondary antibody (1:1000, Jackson ImmunoResearch, West Grove, PA, USA. Cat. No. 111-585-045) (Appendix A). To assess the long-term effects of stress and behavioral control on local neuronal density, brain slices containing dorsal DG (Bregma: −1.34–−2.30 mm) were stained by primary rabbit anti-NeuN (1:500, Millipore, Temecula, CA, USA, Cat. No. MABN140) (Appendix A) and incubated with a FITC conjugated (1:500, Jackson ImmunoResearch, West Grave, PA, USA, Cat. No. 111-545-003) secondary antibody (Appendix A). Apoptotic cells were visualized by TUNEL (TdT-mediated dUTP nick-end labeling) Assay Kit-HRP-DAB (Abcam, Cambridge, UK, Cat. No. ab206386) according to the manufacturer’s instructions. Briefly, brain slices were incubated with proteinase K at room temperature for 10 min. Endogenous peroxidase was inactivated by 3% H_2_O_2_/methanol for 5 min. The slices were incubated with a TUNEL reaction mixture in a humidified chamber at 37 °C for 90 min, followed by washing with 1X TBS (Tris-buffered saline). The TUNEL reaction mixture consisted of an enzyme solution (terminal deoxynucleotidyl transferase) and a nucleotide mixture. The slices were incubated with a converter anti-fluorescein antibody conjugated with anti-FITC horseradish peroxidase (HRP) at 37 °C for 30 min. After washing with TBS, the immunoreaction was visualized by incubating with 3,3-diaminobenzidine tetrahydrochloride (DAB) in H_2_O_2_. Sections were counterstained with methylene green and incubated for 3 min. Slices were mounted with Entellan^TM^ rapid mounting medium (Merck KGaA, Darmstadt, Germany, Cat. No. 107961) for imaging. To assess the long-term effects of stress and behavioral control on local autophagic flux, brain slices were homogenized in an ice-cold lysis buffer containing a protease inhibitor cocktail (Roche, Basel, Switzerland, Cat. No. 11873580001). The samples were centrifuged at 13,500× *g* for 10 min at 4 °C, and the protein concentrations of supernatants were determined using the Bradford method (BioRad Laboratories, Hercules, CA, USA, Cat. No. 500-0006). Equal quantities of protein (20 μg) from each sample were re-suspended in the loading buffer, denatured at 70 °C for 10 minutes, and loaded into the wells of the sodium dodecyl sulfate polyacrylamide gel electrophoresis. After electrophoresis, the protein was transferred onto polyvinylidene fluoride membranes (Thermo Fisher Scientific, Waltham, MA, USA, Cat. No. 88518). The membranes were blocked with a nonfat dry milk buffer (5%) for 1 hour and subsequently incubated overnight at 4 °C with the following primary antibodies: rabbit anti-LC3A/B (1:1000, Cell Signaling Technology, Danvers, MA, USA, Cat. No. 4108S), mouse anti-p62 (1:1000, Abcam, Cambridge, MA, USA, Cat. No. ab56416), and mouse anti-β-actin (1:10000, Merck Millipore, Middlesex County, MA, USA, Cat. No. MAB1501) (Appendix A). The membranes were then processed using the secondary anti-rabbit antibody (1:1000, Jackson ImmunoResearch Laboratories, West Grove, PA, USA, Cat. No. 111-035-003) and anti-mouse antibody (1:5000, Jackson ImmunoResearch Laboratories, West Grove, PA, USA, Cat. No. 515-035-003) (Appendix A) for 1.5 h at room temperature. The blots were displayed with an ECL^TM^ Western blot detection kit (Perkin-Elmer^TM^ Life Sciences, Boston, MA, USA, Cat. No. PK-NEL105001EA).

### 4.4. Kainic Acid-Induced Seizure and Triple Staining for Dorsal DG BrdU/NeuN/Arc-Expressing Neurons

Seizure-induced immediate-early gene, Arc, expression has been used for labeling adult-born, circuit-incorporated neurons, primarily due to synapse and glutamate receptor formation [68]. An intraperitoneal (i.p.) kainic acid injection (10 mg/kg; Sigma, St. Louis, MO, USA. Cat. No. K0250) was given to induce mice’s tonic and clonic seizures [69]. Such kainic acid treatment may strongly stimulate dorsal DG BrdU-labeled neurons via epileptic activation onto their newly-formed synapses with other extant ones [26]. Approximately 25–35 min after the kainic acid injection, mice exhibited initial seizure activity by exhibiting head nodding, followed by forelimb clonus, and progressively to a seizure stage, characterized by repeated rearing and falling episodes. An i.p. sodium pentobarbital (50 mg/kg, SCI Pharmtech, Inc., Taoyuan, Taiwan, Cat. No. 051100) was given 30 min after the onset of the stage 5 seizure [70] to prevent further seizures. Animals were then perfused 60 min after the injection. Their brains were removed and postfixed in a 4% paraformaldehyde solution overnight at 4 °C and subsequently cryoprotected in a 30% sucrose solution for 48 h at 4 °C. Coronal sections, at 20 μm thickness, were made using a microtome (Thermo Fisher Scientific, Cleveland, OH, USA, Model: CryoStar NX50 OP). Triple staining for BrdU, NeuN, and Arc was conducted using mouse anti-BrdU (1:500, Millipore, Temecula, CA, USA, Cat. No. B5002), rabbit anti-Arc (1:1000, Synaptic Systems, Goettingen, Germany, Cat. No. 156 003), and chicken anti-NeuN (1:100, Millipore, Temecula, CA, USA, Cat. No. ABN91) primary antibodies (Appendix A) in blocking buffer, and incubated with Alexa Fluor 488-conjugated sheep anti-mouse (1:500, Jackson ImmunoResearch, West Grove, PA, USA, Cat. No. 515-545-062), Alexa Fluor 594-conjugated goat anti-rabbit (1:1000, Jackson ImmunoResearch, West Grove, PA, USA, Cat. No. 111-585-045), and CF350 conjugated donkey anti-chicken (1:100, Sigma, St. Louis, MO, USA, Cat. No. SAB4600219) secondary antibodies (Appendix A) in PBS and imaged with an Olympus fluorescent microscope (Olympus, Tokyo, Japan, Model: IX71).

### 4.5. Operant Conditioning and Running

Learning may enhance adult DG neurogenesis [71]. To parcel out learning effects in our stress escapable/inescapable paradigm, an appetitive operant conditioning method was employed. That is, sucrose pellet-supported lever pressing was performed using commercial operant chambers designed for mice, as previously described [72]. In brief, mouse body weight was first reduced to 95% of the original one using our mild food deprivation protocol [72]. The mice then received sucrose pellets in two hooked chambers, where mouse learners obtained sucrose pellets by pressing the active lever under a fixed ratio 1 schedule in one chamber, and their respective non-learner yokes received real-time sucrose pellet delivery in another with both levers extracted. A maximum of 60 pellets were available for each training day. Both learners and non-learner yokes received 7 consecutive days of BrdU injection (100 mg/kg/day), starting from the fourth day of this 10-day training regimen. Some learners and non-learners were only used for serum CORT assays 30 min after the conclusion of the fifth day of training.

Likewise, ES, but not IS, mice may have escalated neurogenesis due to their running, i.e., operant darting behavior, effects [48,73,74]. To best exclude the impact of such running on ES mice’s neurogenesis escalation, manually-elicited darting was used. Mice (runners) received a 10-day tail base-grasping attempt (60 times/1 h/day), while the paired counterparts (non-runner yokes) received no such attempt in the same footshock delivery chambers. The tail base-grasping attempt was also “pseudo-randomly” scheduled as the footshock onset schedule in the footshock stressor regimen. Likewise, both runners and non-runner yokes received 7 doses (100 mg/kg) of BrdU, starting from the fourth day of this forced darting regimen. Moreover, some runners and non-runners were used for serum CORT assays 30 min after the conclusion of the fifth day of training.

### 4.6. Serum Corticosterone (CORT) Level

To assess animals’ stress levels in response to footshock stressors, mice’s whole blood was collected 30 min after the conclusion of the 5th and 10th day of the footshock regimen. Likewise, stress levels of learner/non-learner and runner/non-runner mice were gauged 30 min after the conclusion of the 5th day’s session of their regimens. Mice were killed through rapid decapitation, and their trunk blood was collected in vials sitting at room temperature for approximately 20 min. Blood samples were then centrifuged at 4 °C for 10 min (1000 g), and serum was obtained and immediately frozen (−80 °C) until assay. Serum CORT concentrations were determined using a CORT enzyme-linked immune-sorbent assay kit (Cayman Chemical Co, Ann Arbor, MI, USA, Cat. No. K014-H1), according to the manufacturer’s protocol and an ELISA reader (MULTISKAN EX, Thermo Electron Corp., Finland, Cat. No. 51118177), was used. The intra-assay variability was 7.8%.

### 4.7. Object Location Task Versions and Temozolomide (TMZ) Treatment

The object location task consisted of four sessions, including a 10 min habituation [free navigation in arenas (chambers)], 5 min training (exploration for two identical objects), 5 min object location retention (animals’ return to their holding cages), and a 5 min test session (exploration for two same objects with only one being relocated), all in a dimly lit test room. For acclimation purposes, mice were allowed to navigate in empty arenas in the habituation session, while free to explore two newly introduced identical objects being placed in the same-color, opposite sides, or corners in the training session [37]. The time mice spent exploring the two objects was recorded, and the object with a lesser time spent for exploration was removed to a different-color side or corner during retention and started the test session. The recognition percentage, i.e., the ratio of the time spent exploring the object being relocated over the time spent exploring both objects, was used to determine the quality of the object location working memory. It is important to note that arenas and objects were thoroughly cleaned with 70% ethanol between training and test sessions to prevent remnant olfactory cues. Four versions of the object location task were used (parallelogram arena (chamber) with a 36 cm side, ceramic cup objects with a diameter of 6.3 cm; regular pentagon arena (chamber) with a side length of 27 cm, ceramic cup objects with a diameter of 6.2 cm; regular hexagon arena (chamber) with a side length of 20 cm, ceramic cup objects with a diameter of 4.0 cm; rectangle (40 cm × 41 cm) arena (chamber), ceramic cup objects with a bottom diameter of 8.5 cm) (Appendix A). Systematic between-group counterbalancing of these four versions was conducted so that the same number (one-fourth) of mice from each group started with one of the four versions. Three groups (ES, IS, and control) of mice received these 4 versions of the task 3 days, 3 weeks, and 6 weeks after the conclusion of the stressor regimen. Since ES mice were found to outperform IS and control mice in this object location task at 6 weeks after the stressor regimen, these three groups were further subjected to such tasks for another three consecutive days. Thus, an individual mouse from each group also received those 4 versions in a systematic within-group counterbalancing manner during these four days of the trial.

In a pilot study, ES mice received three consecutive days of treatment (25 mg/kg/day) with temozolomide (TMZ) (Sigma, St. Louis, MO, USA. Cat. No. T2577), an antimitotic drug, one hour after the completion of the daily footshock session during days 4–6 of the stressor regimen. Further, their respective IS and control mice received equivalent volumes of vehicle (saline) injections at the same time points (Appendix A). This TMZ treatment protocol was found to produce an ostensible reduction in mitotic completion in ES mice. Further, ES mice had an indistinct number of BrdU-labeled cells in the dorsal DG compared to control and IS mice 3 days after the conclusion of the 10-day stressor regimen (Appendix A). Thus, the TMZ treatment protocol was used in ES mice to experimentally corroborate the correlational findings regarding their escalations in both the number of 6-week-old, adult-born mature DG neurons and object location memory. Likewise, this TMZ protocol was employed in IS mice, as well, in an attempt to test the hypothesis that a minimal number of 6-week-old, adult-born mature DG neurons may play an essential role in warranting the intactness of object location working memory.

It is important to note that mice were not repeatedly used for variant bioassays and behavioral tasks. However, dorsal DG (BrdU/NeuN)-positive numbers were gauged in mice one day after the conclusion of their object location task in the TMZ experiment.

### 4.8. Statistical Analysis

All analyses were conducted by employing Prism 7 (GraphPad Software, Boston, MA, USA). One-way ANOVAs were employed to assess group differences in dorsal and ventral DG BrdU/DCX-positive cell counts followed by Bonferroni’s post hoc comparisons, if appropriate. Likewise, one-way ANOVAs were used to assess group differences in dorsal DG BrdU/NeuN- and BrdU/NeuN/Arc-positive cell count results followed by Bonferroni’s post hoc comparisons, if appropriate. One-way ANOVAs were adopted to reveal group differences in the long-term effects of the stressor regimen, with and without behavioral control, on dorsal DG mitosis, neuronal density, and autophagic flux. Group differences in dorsal DG apoptosis were analyzed using Kruskal–Wallis tests followed by Dunn multiple comparisons, if appropriate. One-way ANOVAs were completed to reveal plausible group differences in recognition ratios of object location tasks 3 days and 3 weeks after the conclusion of the stressor regimen. Two-way ANOVAs (repeated measure x groups) were used to unveil four consecutive days (sessions) of recognition ratio differences in the object location task 6 weeks after the conclusion of the stressor regimen. Likewise, two-way ANOVAs were used to analyze the recognition ratio and time spent in object exploration in the TMZ experiments. However, one-way ANOVAs were completed to reveal group differences in BrdU/NeuN-positive cells in dorsal DG one day after the conclusion of the 2 days of object location tasks in the TMZ experiment. Mouse serum CORT levels were analyzed by Kruskal–Wallis tests followed by Dunn multiple comparisons for the ES and IS experiments. However, paired t-tests were employed to analyze serum CORT differences in learner vs. non-learner and runner vs. non-runner yokes. One-way repeated measure ANOVAs were used to analyze the acquisition curves of mice’s operant darting and level-pressing behavior followed by Bonferroni’s post hoc tests, if appropriate. The level of statistical significance was set at *p* < 0.05.

## 5. Conclusions

Using the stress escapable/inescapable paradigm, we first conclude that repeated stressors do not necessarily affect adult neurogenesis in the dorsal dentate gyrus, although these repeated stressors persistently exhibit a corticosterone secretion-stimulating effect. Importantly, stressed animals having behavioral control may display adult neurogenesis escalation in the dorsal dentate gyrus. Further, a greater number of 6-week-old and functionally-integrated neurons in the dorsal dentate gyrus may facilitate animals’ spatial working memory performance. Finally, adult-born, 6-week-old neurons in the dorsal dentate gyrus seem to contribute spatial location working memory in a use-dependent manner.

## Figures and Tables

**Figure 1 ijms-24-01983-f001:**
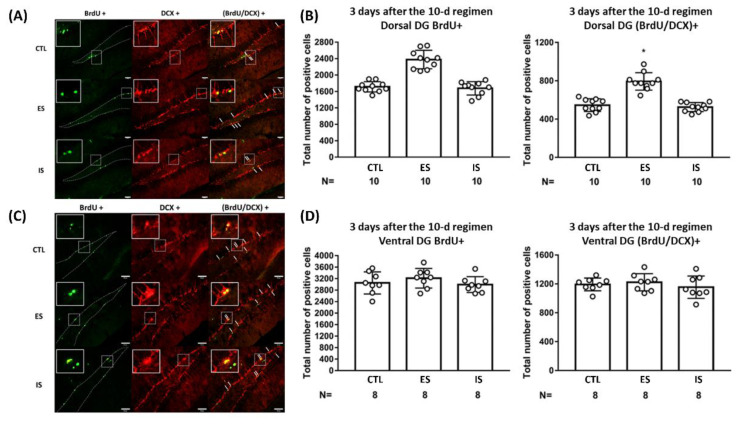
The impact of the footshock stressor and behavioral control on cell proliferation and early neurogenesis in the dorsal and ventral dentate gyrus (DG). (**A**) Representative photomicrographs of BrdU (green)-, DCX (red)-, and BrdU/DCX (yellow)-positive cells were listed in mouse dorsal DG. Staining cells were counted positive only when their locations were at the border (dotted lines) of suprapyramidal and infrapyramidal blades with consistent morphological standards [37]. White arrows point to (BrdU/DCX)-positive cells. Scale bar = 50 µm. (**B**) ES mice had a greater number of BrdU+ and (BrdU/DCX)+ cells compared to their respective IS and control (CTL) mice in the dorsal DG. * Significantly higher than the other groups. Histogram data are plotted as mean ± SD. (**C**) Representative photomicrographs of BrdU (green)-, DCX (red)-, and BrdU/DCX (yellow)-positive cells were listed in the mouse ventral DG. Staining-positive cells were counted using the aforementioned standards. White arrows point to (BrdU/DCX)-positive cells. Scale bar = 50 µm. (**D**) Three groups of mice had an indistinct number of BrdU+ and (BrdU/DCX)+ cells in the ventral DG. Data are plotted as mean ± SD.

**Figure 2 ijms-24-01983-f002:**
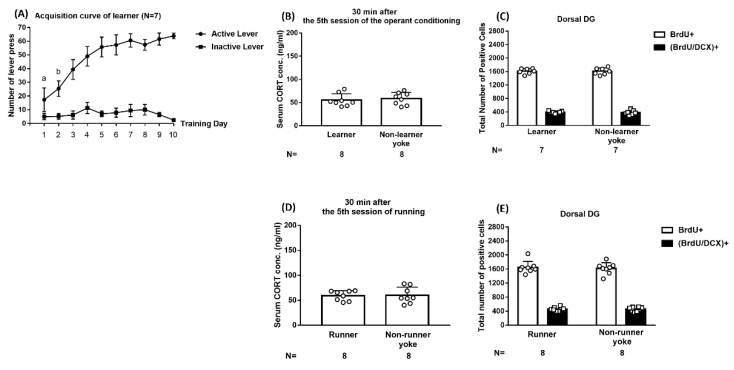
Impact of operant conditioning and forced running on corticosterone (CORT) secretion and short-term neurogenesis in the dorsal DG. (**A**) Mouse learners had comparable lever-pressing numbers on days 3–10 of the regimen. ^a^ Significantly lower than the remaining 9 days. ^b^ Significantly lower than days 3–10. (**B**) Learner and non-learner yoke mice had comparable serum CORT levels 30 min after the conclusion of the 5th day’s training session. (**C**) Learner and non-learner yoke mice had comparable numbers of BrdU+ and (BrdU/DCX)+ cells in the dorsal DG. (**D**) Runner and non-runner yoke mice had comparable serum CORT levels 30 min after the conclusion of the 5th day of the running session. (**E**) Runner and non-runner yoke mice had indistinct numbers of BrdU+ and (BrdU/DCX)+ cells in the dorsal DG. Data are plotted as mean ± SD.

**Figure 3 ijms-24-01983-f003:**
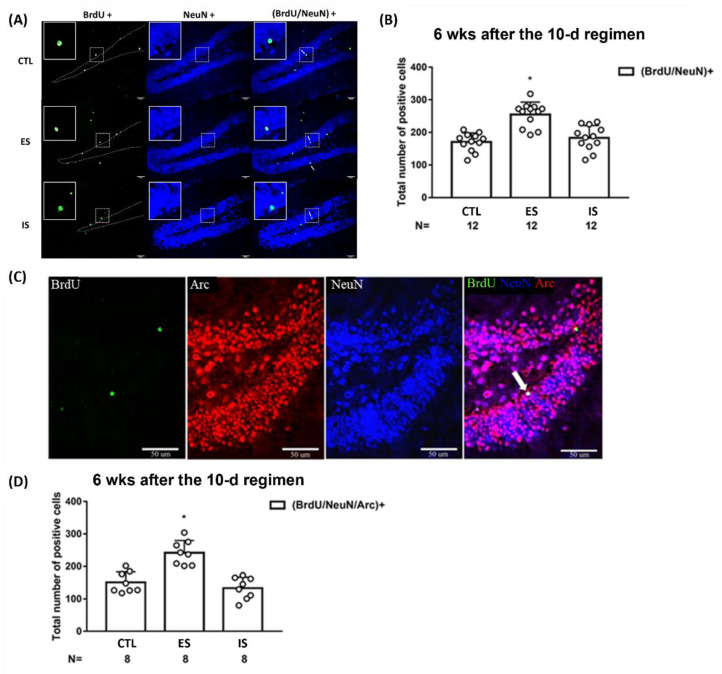
Maturation and functional incorporation of the BrdU-labeled cells in the dorsal DG at 6 weeks after the conclusion of the 10-day footshock stressor regimen. (**A**) Representative photomicrographs of BrdU (green)-, NeuN (blue)-, and BrdU/NeuN (cyan)-positive cells were listed in the mouse dorsal DG. Staining cells were counted positive only when they were near to, not along, the border (dotted lines) of blades. White arrows point to (BrdU/NeuN)-positive cells. All scale bars = 50 µm. (**B**) Group differences in the number of the BrdU/NeuN-positive neurons in the DG. * Significantly greater than the control (CTL) and IS groups. Data are plotted as mean ± SD. (**C**) Representative photomicrographs of BrdU (green)-, Arc (red)-, NeuN (blue)-, and (BrdU/Arc/NeuN) (white)-positive cells in the mouse dorsal DG. A white arrow points to a (BrdU/Arc/NeuN)-positive cell. (**D**) Group differences in the number of the (BrdU/Arc/NeuN)+ neurons in the dorsal DG. * Significantly greater than the control and IS groups. Data are plotted as mean ± SD.

**Figure 4 ijms-24-01983-f004:**
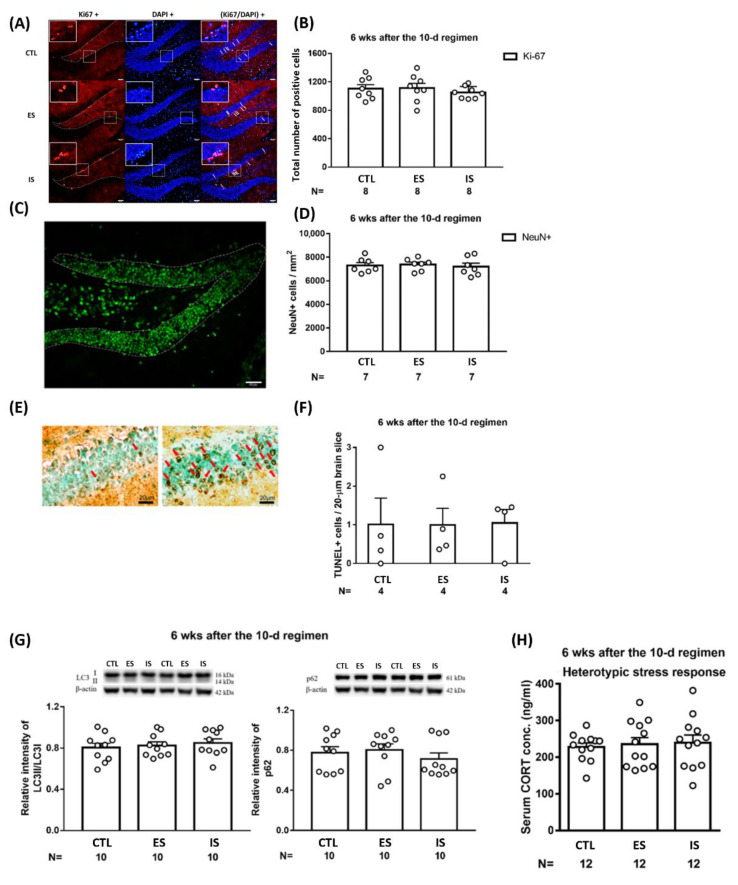
Long-term impact of the 10-day footshock stressor regimen on dorsal DG mitosis, neuronal density, autophagic flux; cells undergoing apoptotic death; and animals’ heterotypic stress-provoked CORT secretion. (**A**) Representative photomicrographs of Ki67 (red)-, DAPI (blue for depicting the border of pyramidal blades)-positive cells. Staining cells were counted positive only when their locations were at the border (dotted lines) of suprapyramidal and infrapyramidal blades. White arrows point to Ki67-positive cells. Scale bar = 50 µm. (**B**) ES, IS, and control (CTL) mice demonstrated comparable mitosis in the dorsal DG at 6 weeks after the conclusion of the 10-day regimen. Data are plotted as mean ± SEM. (**C**) A representative photomicrograph of NeuN (green)-positive cells and dotted line-encircled area for gauging neuronal density. Scale bar = 50 µm. (**D**) Three groups of mice had comparable neuronal density in the dorsal DG at 6 weeks after the conclusion of the 10-day regimen. Data are plotted as mean ± SEM. (**E**) Representative photomicrographs showing apoptotic cells (dark brown) in the dorsal DG of ES and kainic acid-infused (positive control) mice. (**F**) The three groups of mice had indistinct numbers of cells undergoing apoptotic death in the dorsal DG at 6 weeks after the conclusion of the 10-day regimen. Data are plotted as mean ± SEM. (**G**) The three groups of mice had comparable LC3II/LC3I ratios and p62 levels in tissues containing dorsal DG. Data are plotted as mean ± SEM. (**H**) The three groups of mice had comparable novel, stressor-provoked CORT secretions at 6 weeks after the conclusion of the 10-day regimen. The novel stressor consisted of exposure to an elevated circular platform (diameter = 1 m) under a 1000-lux illumination and 70 dB of white noise for a total of 35 min. Data are plotted as mean + SEM. CORT is a short form of corticosterone.

**Figure 5 ijms-24-01983-f005:**
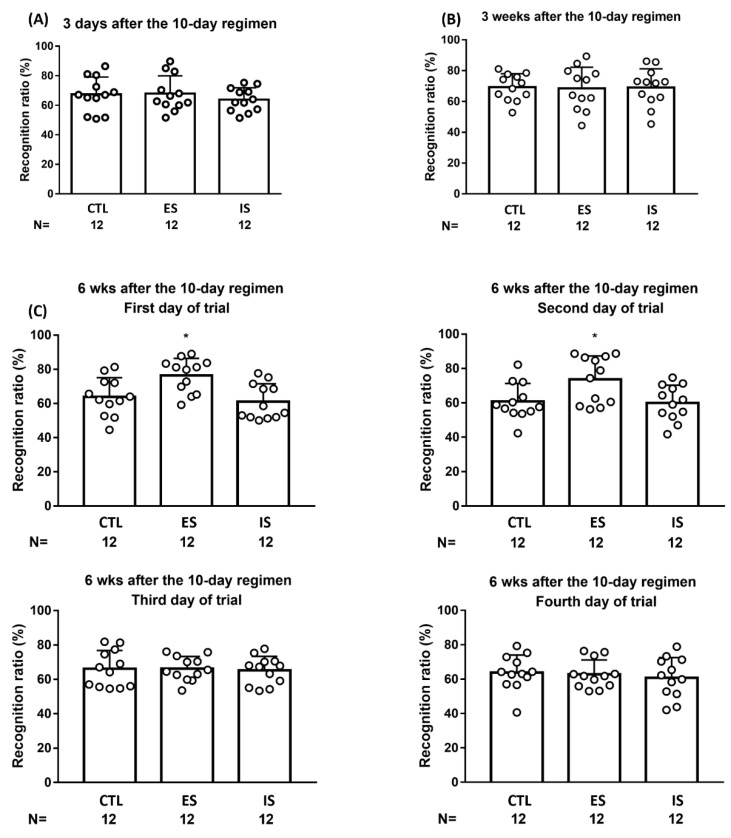
Mouse performances in object location working memory at 3 days, 3 weeks, and 6 weeks after the conclusion of the 10-day stressor regimen. (**A**) ES, IS, and control (CTL) mice performed indistinctly in their recognition ratios 3 days after the conclusion of the regimen. (**B**) ES, IS, and control mice performed comparable recognition ratios 3 weeks after the stressor regimen. (**C**) ES mice outperformed IS and control mice in recognition ratios in this object location task during the first and second, but not the third or fourth, days of the trial 6 weeks after the conclusion of the 10-day stressor regimen. * Significantly greater than IS and control groups. Data are plotted as mean ± SD.

**Figure 6 ijms-24-01983-f006:**
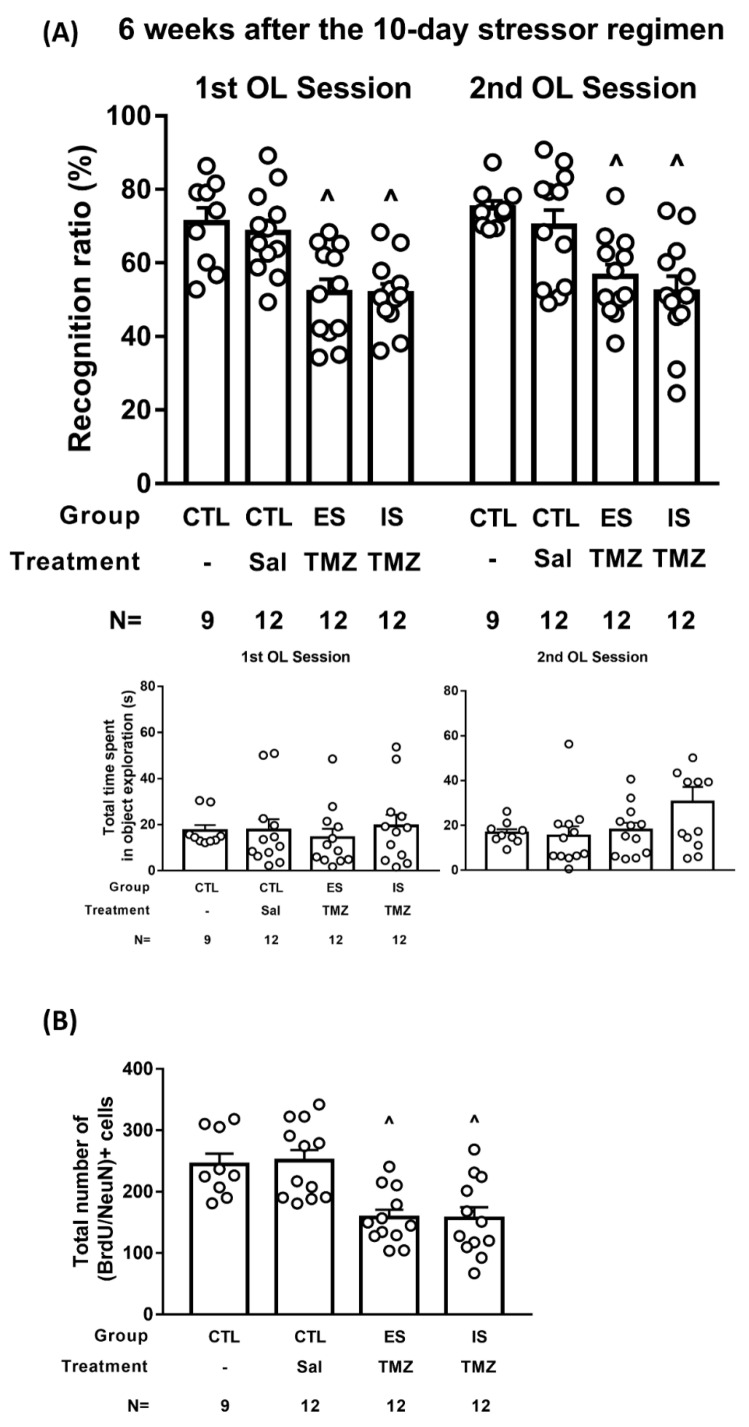
TMZ treatment and object location memory at 6 weeks after the conclusion of the 10-day stressor regimen. (**A**) ES–IS dyads received three daily TMZ injections (25 mg/kg each), while non-stressed control (CTL) mice received an equivalent amount of saline injections or no injection on days 4–6 of the 10-day stressor regimen. ES and IS mice performed comparable recognition ratios on two consecutive days of the trial, and both were lower than the non-stressed controls receiving saline or no injections. ^ Significantly lower than non-stressed groups receiving saline or no injections. Four groups of mice demonstrated indistinct amounts of time spent in object exploration. (**B**) TMZ-treated ES and IS mice had a lower number of dorsal DG (BrdU/NeuN)+ cells than non-stressed control mice. ^ Significantly lower than the non-stressed groups.

**Figure 7 ijms-24-01983-f007:**
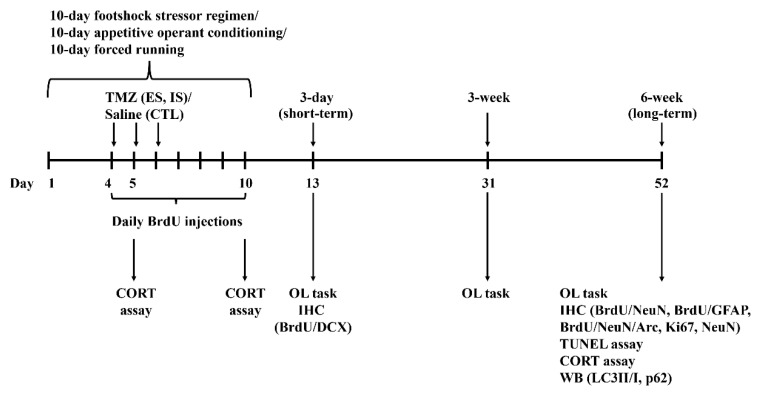
Experimental timelines for the 10-day regimens, biochemical assays, and behavioral tasks. Arc, BrdU, CORT, DCX, GFAP, IHC, NeuN, OL, TMZ, and WB are short forms of activity-regulated cytoskeleton-associated protein, bromodeoxyuridine, corticosterone, doublecortin, glial fibrillary acidic protein, immunohistochemistry, neuronal nuclei, object localization, temozolomide, Western immunoblotting, respectively.

## Data Availability

The raw data supporting the conclusions of this article will be made available by the authors, without undue reservation.

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
