# Peer review of "Male Stressed Mice Having Behavioral Control Exhibit Escalations in Dorsal Dentate Adult-Born Neurons and Spatial Memory"

_ijms, 2023, doi:10.3390/ijms24031983_

Round 1

Reviewer 1 Report

In the present MANU the authors clearly demonstrated that voluntary avoidance of stress (ES mice) led to rapid increase in proliferation of GD cells and that these cells survive, differentiate into neurons and integrate in the DG network in long-lasting manner. However, some questions remain.

1. The authors demonstrated equal level of apoptosis and autophagy between the groups 6 weeks after the stressor regimen. Indeed, both apoptosis and autophagy affect neurogenesis, but in the case of stress these processes develop rapidly, but quickly fade if the stressor ceases to act. It would be interesting to analyze apoptosis/autophagy immediately or a few days after stressor regimen, may be in ES mice new-born cells survive due to low rate of apoptosis/autophagy.

2. When using TMZ treatment, data demonstrated that in a week after the last injection there was no difference in the number of BrdU cells between groups (Suppl Fig 4). However, at 6 weeks thereafter, a decrease in BrdU cells was observed in both ES and IS mice. How can you explain this fact?

3. Why seizure expression by kainic acid was done? There is no clear explanation in either the Methods or the Discussion. In the granular cells Arc is expressed in cell bodies and dendrites as a result of status epilepticus or different behavioral tasks. In Fig 3C, Arc staining is clearly observed in the granular cell layer (cell bodies) and in some hilar cells. In the Discussion the authors wrote (lines 363-365): “Nonetheless, our BrdU/NeuN/Arc-staining findings demonstrated most, if not all, BrdU-labelled mature neurons seemed to be synapsed onto by entorhinal cortex (EC) afferents at this time point [36]”. However, according presented images (Fig 3C) it is not possible to talk about synapses, the magnification should be much larger and be performed by a confocal or electron microscope. Moreover, new-born granular cells form connection/circuits with CA pyramidal cell, and in the cited paper [36] Miller and Sahay wrote about formation of circuits between new-born granular DG cell and CA3.

Minor comments

1. Please clarify in the Methods how many mice were used for each group/analysis

2. the quality of immunostaining images is very low

Author Response

Reviewer #1:

  1. The authors demonstrated equal level of apoptosis and autophagy between the groups 6 weeks after the stressor regimen. Indeed, both apoptosis and autophagy affect neurogenesis, but in the case of stress these processes develop rapidly, but quickly fade if the stressor ceases to act. It would be interesting to analyze apoptosis/autophagy immediately or a few days after stressor regimen, may be in ES mice new-born cells survive due to low rate of apoptosis/autophagy.

Our Responses: We absolutely appreciate Reviewer#1 proposal, especially the experiment on gauging apoptosis and autophagy a few days after the conclusion of the stressor regimen. Our unpublished data indicates that ES mice had greater number of BrdU-labelled neuroblasts in dorsal DG as compared to IS and controls 4 weeks after the conclusion of the stressor regimen. Thus, apoptosis and autophagy assessment prior to this time point (4 weeks), perhaps not “immediately” after the last round of the footshock, will be of significance to identify whether maintaining low rate of apoptosis and autophagy in ES mice may favor their neuroblast and/or neuron survival. In our experimental design, we had apoptosis and autophagy examined at 6 weeks after the conclusion of the stressor/behavioral control regimen. Using such design, dorsal DG environmental niche, at least apoptosis and autophagy, did not seem to contribute to ES mice’ outperformance in object location working memory.

  1. When using TMZ treatment, data demonstrated that in a week after the last injection there was no difference in the number of BrdU cells between groups (Suppl Fig 4). However, at 6 weeks thereafter, a decrease in BrdU cells was observed in both ES and IS mice. How can you explain this fact?

Our Responses: In Suppl. Fig. 4, we used a pilot study and reported a feasible TMZ treatment protocol to effectively reduce ES’ BrdU/DCX-positive neuroblast to a level comparable to IS and control’s three days after the conclusion of the stressor regimen. To vigorously test the hypothesis that ES’ greater number of BrdU-labelled and functionally integrated neurons in dorsal DG contribute to their best object location memory performance and lower number of these adult-born neurons may compromise such memory performance, both ES and IS mice received the TMZ protocol in formal experiment (Fig. 6). Six weeks after the conclusion of the stressor regimen, both ES and IS mice showed lower number of dorsal DG BrdU/NeuN-labelled neuron as compared to the saline-treated controls. Likewise, these ES and IS mice had poor performance in object location working memory but saline-treated controls performed reliable object location working memory. In support of our hypothesis, number of 6-weeks-old and functionally-integrated neurons in dorsal DG indeed affected the quality of object location working memory. TMZ-treated ES and IS mice had paradoxically comparable dorsal DG BrdU/NeuN-labelled numbers and both lower than saline-treated controls 6 weeks after the stressor regimen. In addition to its anti-mitotic effects, the left-over effects of TMZ were suspected to further debilitate the survival and maturation of the neuroblast in this regard.

  1. Why seizure expression by kainic acid was done? There is no clear explanation in either the Methods or the Discussion. In the granular cells Arc is expressed in cell bodies and dendrites as a result of status epilepticus or different behavioral tasks. In Fig 3C, Arc staining is clearly observed in the granular cell layer (cell bodies) and in some hilar cells. In the Discussion the authors wrote (lines 363-365): “Nonetheless, our BrdU/NeuN/Arc-staining findings demonstrated most, if not all, BrdU-labelled mature neurons seemed to be synapsed onto by entorhinal cortex (EC) afferents at this time point [36]”. However, according presented images (Fig 3C) it is not possible to talk about synapses, the magnification should be much larger and be performed by a confocal or electron microscope. Moreover, new-born granular cells form connection/circuits with CA pyramidal cell, and in the cited paper [36] Miller and Sahay wrote about formation of circuits between new-born granular DG cell and CA3.

Our Responses: A paragraph has been added in Materials and methods section for rationale establishment of kainic acid, tonic-clonic seizure and Arc expression. In Discussion section, we rewrite the paragraph to best encompass afferent sources. Likewise, z-stack microscopic methods have been added in Materials and methods section.

Minor comments

  1. Please clarify in the Methods how many mice were used for each group/analysis

Our Responses: Number of mice used for each experiment has been added as now seen in the Results section.

  1. the quality of immunostaining images is very low

Our Responses: Per our sampling range (bregma: -1.34 to -2.30), dorsal DG sub-granular layer indeed comprises a large volume. To fairly reveal inter-group differences on dorsal DG neurogenesis, at least 8 mice were used for each group. Thus, fluorescent images were processed by fluorescent microscopy. To best avoid our imaging/counting bias, z-stack methods were used. That is, a composite z-stack image was first generated by composing 3-4 optical sections for each 20-μm slice. Fluorescence-positive cells on each composite image were then counted by a rater blind to the grouping.

Reviewer 2 Report

There are many tricky sentences in the manuscript, and the paragraphs need to be better related to each other. 

The authors should reorganize the manuscript to clarify the concept behind the paper. They could conclude each chapter with some conclusions and try to link all the endings. The methods section should describe only the experimental procedure and not the rationale (i.e., lines 446-452 or 570 to 572 or 602 to 607, etc.). Adding an antibody table can help to understand the methods section better. 

The authors should be more accurate in the text deleting a few mistakes and repetitions (i.e., 676).

Try to summarise the conclusion in a shorter version and more available to the reader. In my opinion, it's not too clear why the authors chose to use the TMZ treatment. The author should add CTR-TMZ treatment. 

Author Response

Reviewer #2:

1.There are many tricky sentences in the manuscript, and the paragraphs need to be better related to each other.

The authors should reorganize the manuscript to clarify the concept behind the paper. They could conclude each chapter with some conclusions and try to link all the endings.

Our Responses: We have requested a native English scholar to assist us editing and proofing our revised manuscript. As Reviewer#2’s suggestion, concluding remarks have been made for each paragraph in Results section.

2.The methods section should describe only the experimental procedure and not the rationale (i.e., lines 446-452 or 570 to 572 or 602 to 607, etc.). Adding an antibody table can help to understand the methods section better.

Our Responses: Those paragraphs have been truncated or removed. A table (Table 1) has been added to summarize the antibodies used as now seen in Materials and methods section.

3.The authors should be more accurate in the text deleting a few mistakes and repetitions (i.e., 676).

Our Responses: Reviewer#2’s reminders are appreciated. Typos and mistakes have been corrected.

4.Try to summarise the conclusion in a shorter version and more available to the reader. In my opinion, it's not too clear why the authors chose to use the TMZ treatment. The author should add CTR-TMZ treatment.

Our Responses: The Conclusions section has been re-written. To vigorously test the hypothesis that ES’ greater number of BrdU-labelled and functionally integrated neurons in dorsal DG contribute to their best object location memory performance and lower number of these adult-born neurons may compromise such memory performance, both ES and IS mice received the TMZ protocol in our experiment. Six weeks after the conclusion of the stressor regimen, both ES and IS mice showed lower number of dorsal DG BrdU/NeuN-labelled neuron as compared to the saline-treated controls. Likewise, these ES and IS mice had poor performance in object location working memory but saline-treated controls performed reliable object location working memory. In support of our hypothesis, number of 6-weeks-old and functionally-integrated neurons in dorsal DG indeed affected the quality of object location working memory. TMZ-treated ES and IS mice had paradoxically comparable numbers of dorsal DG BrdU/NeuN-labelled neurons and both lower than saline-treated controls 6 weeks after the stressor regimen. In addition to its anti-mitotic effects, the left-over effects of TMZ were suspected to further debilitate the survival and maturation of the neuroblast in this regard.

Reviewer 3 Report

In their manuscript Sun et al. examine whether repeated exposure to stress followed by behavioral control had any positive effect on blood corticosterone level, spatial memory performance and dorsal hippocampal neurogenesis both short- and long- term after stress termination. The theme and findings are interesting. However, the experimental design is quite confusing and rationale for the performance of each procedure and treatment is not clearly stated. Some other points that the authors should pay attention to:

1.       The abstract should be written in a more comprehensive way that will emphasize the aim of the present study, the various manipulations used and their scope and findings.

2.       Although the study presents many interesting findings there are so many different manipulations that were performed at various time points that hamper the understanding of the experimental design. I believe that the authors should add a figure/diagram or even better a flow chart of the whole experimental design (with the various manipulations, time points and groups/number of animals used in each time point) in order the manuscript to be more comprehensive.

3.       How many C57Bl/6 male mice were used in total in this study and how were all these mice allocated in the different experimental  groups, procedures and treatments?

4.       All immunostained dorsal hippocampal sections were imaged and counted under a fluorescent microscope? A confocal laser scanning microscope would be more appropriate for that since multiple immunolabeling was performed.  Please discuss on that.

5.       To my opinion, the various antibodies used should be referred as mouse anti-BrdU antibody (Line 501) and not mouse anti-mouse BrdU. The same for rabbit anti-Ki67 antibody (Line 523), rabbit anti-NeuN antibody (Line 529), anti-Arc and anti-NeuN antibody (Line 588-590) etch.

6.       Why were four different types of arena (rectangular, regular hexagon, regular pentagon and parallelogram) used? Do obtained findings add value to the manuscript?

7.       Line 634: please refer to the blood sampling method used i.e from tail vein or elsewhere? Animals sedated or else?

Author Response

Reviewer#3:

  1. The abstract should be written in a more comprehensive way that will emphasize the aim of the present study, the various manipulations used and their scope and findings.

Our Responses: Reviewer#3’s suggestions are appreciated. The abstract has been re-written to incorporate all these concerns. The re-written paragraph is listed as follows.

“ Escapable/inescapable stress paradigm was used to study whether behavioral control and repeated footshock stressors may affect adult neurogenesis and related cognitive function. Male stressed mice having behavioral control (ES) had short-term escalation of dorsal dentate gyrus (DG) neurogenesis, while same stressed mice having no such control had unaltered neurogenesis as compared to control mice receiving no stressors. Paradoxically, ES and IS mice had comparable stress-induced corticosterone elevations throughout the stress regimen. Appetitive operant conditioning and forced running procedures were used to model learning and exercise effects in this escapable/inescapable paradigm. And conditioning and running procedures did not seem to affect mice’ corticosterone and short-term neurogenesis. ES and IS mice did not show noticeable long-term changes in their dorsal DG neurogenesis, gliogenesis, local neuronal density, apoptosis, autophagic flux and heterotypic stress responses. ES mice were found to have greater number of previously labelled and functionally integrated DG neurons as compared to IS and control mice 6 weeks after the conclusion of the stressor regimen. Likewise, ES mice outperformed IS and non-stressed control mice for the first two, but not the remaining two, trials in object location task. Compared to non-stressed controls, temozolomide-treated ES and IS mice having lower number of dorsal DG 6-weeks-old neurons display poor performance in their object location working memory. These results, taken together, prompt us to conclude that repeated stressors, albeit their corticosterone secretion-stimulating effect, do not necessary affect adult dorsal DG neurogenesis. Moreover, stressed animals having behavioral control may display adult neurogenesis escalation in dorsal DG. Furthermore, the number of 6-weeks-old and functionally-integrated neurons in dorsal DG seems to confer the quality of spatial location working memory. Finally, these 6-weeks-old adult-born neurons seem to contribute spatial location memory in a use-dependent manner. “

  1. Although the study presents many interesting findings there are so many different manipulations that were performed at various time points that hamper the understanding of the experimental design. I believe that the authors should add a figure/diagram or even better a flow chart of the whole experimental design (with the various manipulations, time points and groups/number of animals used in each time point) in order the manuscript to be more comprehensive.

Our Responses: Reviewer#3’s suggestion is well taken. In an attempt to efficiently summarize the time points for all experiments, a timeline has been used using a new figure (Figure 7).

  1. How many C57Bl/6 male mice were used in total in this study and how were all these mice allocated in the different experimental groups, procedures and treatments?

Our Responses: A total of 562 mice were used in this study. Number of mice used for each experiment has been added in respective experiments in Results section as now seen in the revised manuscript.

  1. All immunostained dorsal hippocampal sections were imaged and counted under a fluorescent microscope? A confocal laser scanning microscope would be more appropriate for that since multiple immunolabeling was performed. Please discuss on that.

Our Responses: Per our sampling range (bregma:-1.34 to -2.30), dorsal DG indeed comprises a large volume. To fairly reveal inter-group differences on dorsal DG neurogenesis, at least 8 mice were used for each group. Thus, fluorescent images were processed exclusively by fluorescent microscopy. To best avoid our imaging/counting bias, z-stack methods were used. That is, a composite z-stack image was first generated by composing 3-4 optical sections for each 20-μm slice. Fluorescence-positive cells on each composite image were then counted by a rater blind to the grouping.

  1. To my opinion, the various antibodies used should be referred as mouse anti-BrdU antibody (Line 501) and not mouse anti-mouse BrdU. The same for rabbit anti-Ki67 antibody (Line 523), rabbit anti-NeuN antibody (Line 529), anti-Arc and anti-NeuN antibody (Line 588-590) etch.

Our Responses: Changes have been made as Reviewer#3’s suggestion as now seen in the revised manuscript.

  1. Why were four different types of arena (rectangular, regular hexagon, regular pentagon and parallelogram) used? Do obtained findings add value to the manuscript?

Our Responses: To test whether 6-weeks-old and functionally integrated neurons may take part in object location working memory in a use-dependent manner, four consecutive days of object location task were employed. Thus, four versions of arena and objects were used to prevent the likelihood that previous learning/memory effects may affect subsequent learning and performance.

  1. Line 634: please refer to the blood sampling method used i.e from tail vein or elsewhere? Animals sedated or else?

Our Responses: To prevent plausible anesthetics-produced confounds on blood CORT assay, mice were killed by rapid decapitation and their trunk blood was collected. This description is now seen in Serum corticosterone (CORT) level section in the revised manuscript.

Round 2

Reviewer 1 Report

I am satisfied with the authors´ answers and with the modifications of the manuscript.